# Strategic Optimization of the Middle Domain IIIA in RBP-Albumin IIIA-IB Fusion Protein to Enhance Productivity and Thermostability

**DOI:** 10.3390/ijms26010137

**Published:** 2024-12-27

**Authors:** Myungho Sohn, Sanggil Kim, Hyeon Ju Jeong, In Young Ko, Ji Wook Moon, Dowon Lee, Junseo Oh

**Affiliations:** 1New Drug Development Center, Osong Medical Innovation Foundation, Osong 28160, Republic of Korea; sohnho2@kbiohealth.kr (M.S.); ryon2174@kbiohealth.kr (S.K.); hyeonju0602@kbiohealth.kr (H.J.J.); koiy@kbiohealth.kr (I.Y.K.); 2Department of Biomedical Sciences, College of Medicine, Korea University, Seoul 02841, Republic of Korea; mjw6132@korea.ac.kr (J.W.M.); idowon8@korea.ac.kr (D.L.)

**Keywords:** fibrosis, fusion protein, structural rigidity, thermal stability, productivity

## Abstract

The protein therapeutics market, including antibody and fusion proteins, has experienced steady growth over the past decade, underscoring the importance of optimizing amino acid sequences. In our previous study, we developed a fusion protein, R31, which combines retinol-binding protein (RBP) with albumin domains IIIA and IB, linked by a sequence (AAAA), and includes an additional disulfide bond (N227C-V254C) in IIIA. This fusion protein effectively inhibited hepatic stellate cell activation. In this study, we further optimized the sequence. The G176K mutation at the C-terminus of RBP altered the initiation site of the first α-helix in domain IIIA, shifting it from P182 to K176, and promoted polar interactions between K176 and adjacent residues, enhancing the rigidity of the RBP/IIIA interface. The introduction of an additional disulfide bond (V231C/Y250C) connecting helices 3 and 4 in IIIA resulted in a three-fold increase in productivity and a 2 °C improvement in thermal stability compared to R31. Furthermore, combining the G176K mutation with V231C/Y250C further enhanced both productivity and anti-fibrotic activity. These findings suggest that the enhanced stability of domain IIIA, conferred by V231C/Y250C, along with the increased rigidity of the RBP/IIIA interface, optimizes interdomain distance and alignment, facilitating proper protein folding.

## 1. Introduction

Tissue fibrosis is a pathological condition marked by the excessive build-up of extracellular matrix (ECM) components, including collagen and fibronectin, in various organs [1]. It results in the loss of organ function and poses a serious threat to health. Despite its harmful effects and extensive research efforts, effective therapies are still lacking, highlighting the urgent need for the development of novel anti-fibrotic drugs. Myofibroblasts, which are the primary producers of ECM proteins, are rarely found in normal tissue but are prevalent in fibrotic tissue. They originate from multiple sources including organ-specific stellate cells, resident fibroblasts, epithelial–mesenchymal transition (EMT), and fibrocytes [2]. Hepatic stellate cells (HSCs) are located in the perisinusoidal space of the liver, where they usually remain quiescent, comprising about 5–8% of total liver cells and serving as a storage site for approximately 80% of the body’s vitamin A [3]. When exposed to fibrogenic stimuli, quiescent HSCs become activated, leading to phenotypic and functional changes that transform them into a myofibroblast-like phenotype [4]. This activation process is characterized by several key features: the loss of cytoplasmic lipid droplets containing vitamin A (retinol), increased cellular proliferation, and elevated expression of alpha-smooth muscle actin (α-SMA) and ECM proteins. Consequently, the activation of stellate cells, which are found in various tissues including the liver, pancreas, kidneys, and lungs, plays a crucial role in tissue fibrogenesis and presents a promising target for anti-fibrotic treatments [5,6,7].

Albumin is the most abundant protein in plasma (~66 kDa). It consists of three structurally similar domains (I, II, and III), each of which is further divided into two subdomains, A and B [8]. Subdomain A contains six α-helices and turns, while subdomain B includes four α-helices. We previously demonstrated that albumin is expressed in quiescent stellate cells but not in activated ones, although its expression levels are lower compared to those in liver cells [9]. Interestingly, forced expression of albumin in activated stellate cells induced a notable reversal of their phenotype, transitioning them from myofibroblasts back to an early activated state, along with a marked decrease in α-SMA and collagen type I expression. These findings, with subsequent studies, led to the development of a novel recombinant fusion protein that combines retinol-binding protein (RBP) with albumin domains IIIA and IB as an anti-fibrotic agent [10]. Studies have demonstrated that this RBP–albumin fusion protein effectively inhibits stellate cell activation and reduces tissue fibrosis [7,11]. RBP was chosen for its ability to facilitate targeted delivery to stellate cells, as it promotes the cellular uptake of retinol in HSCs through its interaction with the membrane receptor STRA6 [12].

Over the past decade, the share of protein-based drugs in the pharmaceutical market has been consistently increasing, particularly emphasizing the significant potential of fusion protein drugs with diverse functions [13]. The necessity for optimizing amino acid sequences has become more crucial than ever to ensure adequate efficacy, productivity, and stability [14,15]. In our previous research, we modified the amino acid sequence of the fusion protein RBP-IIIA-EVDD-IB into a revised version RBP-IIIA(N227C/V254C)-AAAA-IB (referred to as R31), where the linker sequence between IIIA and IB was altered and an additional disulfide bond was introduced in domain IIIA (see Appendix A) [16]. In the present study, we further optimized the amino acid sequence of R31 and found that introducing the G176K mutation along with an additional disulfide bond in domain IIIA improved productivity, thermal stability, and anti-fibrotic activity, likely due to the increased structural stability of the fusion protein.

## 2. Results

### 2.1. The Productivity of the Fusion Protein R31 Needs to Be Enhanced

The fusion protein R31 (amino acids 1–379) is composed of the RBP domain (amino acids 1–177, corresponding to residues 1–177 in RBP) and the albumin domain IIIA (amino acids 178–291, corresponding to residues 380–493 in albumin) and IB (amino acids 292–379, corresponding to residues 107–194 in albumin), as shown in Figure 1 and Figure 2. It includes a signal peptide at its N-terminus for secretion and is tagged with a His-tag at the C-terminus for purification purposes. To optimize the amino acid sequence of R31 for manufacturing needs, including productivity and thermal stability, AlphaFold2 was used to predict its structure. The predicted model showed that the region linking RBP and IIIA in the fusion protein R31 appears flexible or disordered, as highlighted by a black circle (Figure 1A and Appendix A). After C174, which forms a disulfide bond with C70 in RBP4, the first α-helix of IIIA starts at P182. We hypothesized that the flexibility of the region (D175~E181) increases the conformational heterogeneity of the fusion protein, which may contribute to low productivity. Thus, we decided to modify the amino acid sequence of this linker region. Despite several attempts to increase the rigidity by substituting multiple residues with helix-forming amino acids, none were successful. However, recognizing that G176 and R177 are located four residues away from E180 and E181, we substituted G176 with Lys. This change caused a shift in the α-helix initiation site from P182 to K176, leading to an additional helical turn and a more rigid linker region (Figure 1B and Appendix A). Analysis of the AlphaFold2-predicted structure of R31 using Schrödinger revealed that the G176K mutation created additional polar interactions with the neighboring residues 178, 179, 180, and 181, potentially further stabilizing the linker region (Figure 2). The effect of the G176K mutation on productivity was assessed by transiently transfecting Expi293 cells with plasmids encoding the fusion proteins R31 or R31-G176K. The culture supernatant was then analyzed by Western blotting using an anti-His tag antibody. However, no significant increase in productivity was observed, indicating that this point mutation alone did not enhance expression levels (Figure 3).

### 2.2. The Productivity of R31 Was Improved by Incorporating an Additional Disulfide Bond, V231C/Y250C

The first domain of R31, RBP (1–177), shares almost the entire sequence with RBP (1–183), while the second and third domains, IIIA and IB, are partially derived from albumin. As predicted by AlphaFold2, the structure of the first domain of R31 closely matches that of RBP (Figure 1). We hypothesized that the relatively low productivity of R31 could be partially attributed to the low stability of the downstream albumin domain. As a result, we focused on altering the amino acid sequence within domain IIIA. Domain IIIA of albumin contains four native disulfide bonds, one of which (C437/C448 in albumin, corresponding to C235/C246 in R31) connects α-helices 3 and 4 (Figure 4, marked in red). In a previous study, we showed that introducing a disulfide bond via N227C/V254C (marked in blue), which also links α-helices 3 and 4, significantly enhanced both the productivity and efficacy of the fusion protein [16]. Since the distance between α-helices 3 and 4 slightly increases as they move away from the hinge, we hypothesized that the N227C/V254C disulfide bond might not be sufficient to stabilize the two helices. To address this, we added an additional disulfide bond by replacing V231/Y250 (Figure 4, marked in yellow) with cysteine, positioning it between the native C235/C246 bond and N227C/V254C. We tested whether the combination of the N227C/V254C and V231C/Y250C disulfide bonds would enhance the structural stability and protein yield. Expi293 cells were transiently transfected with plasmids encoding the fusion proteins R31, R31-G176K, R31-V231C/Y250C, and R31-G176K-V231C/Y250C, and the culture supernatant was analyzed by Western blotting. Notably, the inclusion of V231C/Y250C led to a significant increase in productivity (Figure 5).

### 2.3. The Fusion Protein R31-G176K-V231C/Y250C Exhibited Improved Productivity and Thermal Stability

To enhance the productivity, we modified the amino acid sequence of the fusion protein R31, resulting in the variants R31-G176K, R31-V231C/Y250C, and R31-G176K-V231C/Y250C. Following the transient transfection of a plasmid encoding the fusion protein into Expi293 cells, the secreted fusion protein (~44 kDa) was purified using Ni Sepharose, followed by size exclusion chromatography. SDS-PAGE analysis was conducted under both reducing conditions (with a final concentration of 10 mM DTT) and non-reducing conditions, confirming the presence of a band corresponding to the fusion protein (Figure 6A). The thicker band observed with 10 mM DTT is likely due to the altered migration pattern of proteins with disrupted disulfide bonds. The size exclusion chromatography profile showed a single, sharp peak, indicating the homogeneity of the protein samples (Figure 6B). The protein yield assessment revealed that R31-V231C/Y250C produced more than three times the yield of R31 (Table 1). Additionally, combining V231C/Y250C with G176K resulted in a modest further improvement, reaching over 130 mg/L. To evaluate the thermal stability of the R31 variants, a protein thermal shift assay was conducted. The melting temperature of R31-V231C/Y250C and R31-G176K-V231C/Y250C increased by approximately 2 °C compared to R31 (Figure 7). These results suggest that the introduction of the additional disulfide bond (V231C/Y250C) in R31 may enhance the stability of the middle domain IIIA, thereby improving the fusion protein’s productivity and thermal stability. Furthermore, the G176K mutation, which increases rigidity at the RBP/IIIA interface, combined with V231C/Y250C, likely increases the overall structural stability and promotes proper protein folding.

### 2.4. The Fusion Protein R31-G176K-V231C/Y250C Showed Increased Anti-Fibrotic Activity

We then measured the anti-fibrotic effects of the fusion proteins R31, R31-G176K, R31 -V231C/Y250C, and R31-G176K-V231C/Y250C on activated HSCs in vitro, which were isolated from mice as described in the Materials and Methods. After treating HSCs after passage 1 with the purified fusion proteins, we quantified the expression of α-SMA and collagen type I, two well-established markers of activated HSCs/myofibroblasts [17], using real-time PCR. The fusion proteins reduced mRNA expression, with R31-G176K-V231C/Y250C having the most pronounced effect (Figure 8). These results were also accompanied by noticeable phenotypic changes, including the reappearance of lipid droplets, cell shrinkage, and a reduction in stress fibers (Figure 9).

## 3. Discussion

Fusion proteins have gained considerable attention in the drug industry. These proteins combine parts of two or more proteins into one, creating new molecules with specialized functions. A common application is the integration of therapeutic activity with targeting capabilities, which improves specificity and minimizes off-target effects [18]. An example of this approach is the RBP–albumin fusion protein studied here, where albumin serves as the functional domain and RBP acts as the targeting domain.

Optimizing the amino acid sequence of a fusion protein is a critical but challenging task, as each amino acid influences the protein’s stability, activity, and structure [15]. Given the vast number of possible amino acid combinations, predicting how individual changes will influence the protein’s function and conformation is not easy. To identify the optimal sequence, researchers typically rely on computational modeling [19,20] and trial-and-error approaches, which are time-consuming and require specialized expertise [21]. Moreover, there is a limited body of literature focusing specifically on sequence optimization for fusion proteins.

Several factors affect the productivity of fusion proteins, including proper protein folding, stability, solubility, and post-translational modifications. The choice of linker sequence between protein domains is critical for ensuring proper folding and stability [14]. Without suitable linkers, directly fusing protein domains can lead to issues such as misfolding, reduced protein yield, or loss of bioactivity. In this study, we carefully examined the linker region between RBP and IIIA and observed a flexible connection between 175D and 181E. According to George and Heringa [22], rigid linkers often adopt α-helical structures or include multiple proline residues. We substituted G176 with lysine, which shifted the α-helix initiation site from P182 to K176 and increased polar interactions with neighboring residues, thereby enhancing the rigidity of the RBP–IIIA interface. However, this mutation alone did not improve productivity. This could be partly due to the low stability of the downstream albumin domain(s), which likely hinders proper protein folding. On the other hand, in our previous study, we modified the linker between domains IIIA and IB, hypothesizing that the flexibility of the linker caused the fusion protein to adopt a compact conformation. We replaced the sequence (EVDD) with one (AAAA) that is neither flexible nor rigid to ensure proper spacing between the domains. Thus, the choice of linker sequence should be tailored to the specific context of the fusion protein [23].

To enhance the productivity of R31, we introduced an additional disulfide bond through the V231C/Y250C pair, which lies between the native C235/C246 bond and the N227C/V254C pair. Previous studies have shown that the artificial insertion of disulfide bonds can strengthen the protein structure and improve its functionality [24,25]. The inclusion of V231C/Y250C resulted in a notable increase in both productivity and thermal stability, indicating that the enhanced structural stability of the middle domain IIIA, conferred by V231/Y250, facilitates proper protein folding. Moreover, combining V231C/Y250C with the G176K mutation produced a synergistic effect. We believe that the improved stability of IIIA coupled with the increased rigidity of the RBP/IIIA interface optimize the interdomain distance and alignment of IIIA relative to RBP. This adjustment likely contributes to an additional increase in productivity and efficacy. A similar example has been reported, where rigid helical linkers enhanced protein crystallization by ensuring proper domain alignment [26].

We used AlphaFold2 to predict the fusion protein structure in order to understand the low yield of the fusion protein R31 and identify potential solutions. AlphaFold2 provides pLDDT scores and PAE values to assess prediction confidence and model reliability [19]. While these scores offer valuable insights, validation remains essential for ensuring biological relevance and structural consistency [27]. Nevertheless, AlphaFold2 has proven to be an invaluable tool for designing and optimizing amino acid sequences to improve the protein’s stability and productivity.

We evaluated the effects of each amino acid sequence modification on productivity, thermal stability, and anti-fibrotic activity. Compared to R31, the introduction of the additional disulfide bond (V433C/Y452C) and the G176K mutation resulted in a significant increase in both productivity and thermal stability. When combined with the outcomes from our previous sequence optimization efforts [16], the enhanced structural stability of the middle domain IIIA, along with the optimization of both its RBP/IIIA and IIIA/IB interfaces, led to more than a 100-fold increase in fusion protein productivity. This study is the first to highlight the importance of optimizing the middle domain in fusion protein design to enhance both productivity and efficacy. We believe that it offers valuable insights into the complexities of fusion protein production and serves as an example of rational fusion protein design.

## 4. Materials and Methods

### 4.1. Animals

Male BALB/c mice were obtained from Orient Bio, Inc. (Sungnam, Republic of Korea), and housed in a controlled environment with regulated temperature, humidity, and lighting. All animal procedures were approved by our institutional review board (KOREA-2024-0030) and adhered to the NIH Guide for the Care and Use of Laboratory Animals. Expi293 cells were sourced from Thermo Fischer Scientific (Waltham, WA, USA).

### 4.2. Expression and Purification of Fusion Proteins

Fusion proteins, including R31 (depicted in Appendix A) and its variants, were synthesized using a previously established protocol [16]. Briefly, Expi293 cells were transiently transfected with plasmids using the ExpiFectamine 293 transfection reagent (Thermo Fisher Scientific, Waltham, WA, USA) according to the manufacturer’s instructions. Six days post-transfection, the culture medium was clarified by centrifugation. Fusion proteins were then purified by affinity chromatography using a HisTrap Excel column (Cytiva, Marlborough, MA, USA), followed by size exclusion chromatography on a HiLoad Superdex 200 16/60 GL column (Cytiva). The buffer of the R31 variants was exchanged, and proteins were concentrated using 10 kDa molecular weight cutoff centrifugal filter units (Millipore, Burlington, MA, USA).

### 4.3. SDS-PAGE and Western Blot Analysis

Protein samples were prepared by mixing with 4× NuPAGE™ LDS Sample Buffer (Invitrogen #NP0007, Carlsbad, CA, USA), with or without 10× NuPAGE™ Sample Reducing Agent (500 mM DTT, Invitrogen #NP0009), and were then denatured at 100 °C for 5 min. Denatured proteins were separated on 4–12% bis-tris NuPAGE gels (Thermo Fisher Scientific) and subjected to either Coomassie staining or Western blot analysis. A rabbit polyclonal anti-His antibody (Abcam #ab1187, Cambridge, MA, USA) was employed for detection. The Western blot results were quantified using ImageJ (version 1.53m) to analyze band intensities, and the relative expression levels of the proteins were calculated [28,29].

### 4.4. Computational Analysis of Residue Interactions

The 3D structure predicted by AlphaFold2 served as the basis for subsequent calculations and visualizations, which were performed using the Schrödinger Suite Release 2024-2. The structure was prepared with Schrödinger’s Protein Preparation Wizard, during which all crystallographic water molecules were removed to optimize the model for interaction analysis. Residue interaction analysis was conducted using the Prime module [30], focusing on identifying polar interactions that could influence protein stability and function.

### 4.5. Protein Thermal Shift (PTS) Assay

The PTS was performed in MicroAmp™ Fast Optical 96-Well Reaction Plates with Barcode (Applied Biosystems, Carlsbad, CA, USA) using the Applied Biosystems ViiA 7 Real-Time PCR System. Protein samples were centrifuged for 15 min before preparation. The final reaction mixture contained R31 variant proteins and Protein Thermal Shift™ Dye (1000× stock; Applied Biosystems), diluted at a ratio of 1:250 in PBS (pH 7.4). Melting curve data were recorded from 25 °C to 99 °C with an increment rate of 0.05 °C/s. Excitation and emissions filters were applied for Protein Thermal Shift™ Dye (470 and 520 nm, respectively). The resulting fluorescence data were analyzed, and the melting temperatures were calculated using Protein Thermal Shift™ Software v1.3. All samples were tested in triplicate.

### 4.6. Isolation of Mouse Hepatic Stellate Cells (HSCs)

HSCs were isolated from >14-week-old male BALB/c mice as described previously [31]. Briefly, livers were perfused with phosphate-buffered saline (PBS), followed by Gey’s balanced salt solution (GBSS) containing collagenase (0.5 mg/mL; Sigma-Aldrich, St. Louis, MO, USA) and pronase (1 mg/mL; Sigma-Aldrich). After dissection and removal of gall bladders and connective tissues, liver suspensions were further digested with GBSS containing collagenase (0.25 mg/mL), pronase (0.5 mg/mL), and DNase (0.07 mg/mL; MP Biomedicals, Santa Ana, CA, USA) for 12 min at 37 °C. Cells were centrifuged through a 13.4% Nycodenz (Sigma-Aldrich) gradient, and the enriched HSCs were collected from the interface, washed, and cultured in DMEM with 10% fetal bovine serum. HSC purity was confirmed by microscope. Cells were passaged before reaching 70% confluence and used as activated HSCs. The activation status of the HSCs was assessed by increased α-SMA and collagen type I expression and morphological changes.

### 4.7. Quantitative Real-Time PCR

Total RNA was extracted using TRIzol (Ambion, Austin, TX, USA) and reverse-transcribed into cDNA. Real-time PCR was performed on an ABI QuantStudio 3 system, with GAPDH used as the internal control for normalization. The primers for α-SMA were 5′- CCAGCACCATGAAGATCAAG -3′ (forward) and 5′- TGGAAGGTAGACAGCGAAGC -3′ (reverse); for collagen type I, 5′- CGACCTCAAGATGTGCCACT -3′ (forward) and 5′- CTTGGTTAGGGTCGATCCAG -3′ (reverse); and for GAPDH, 5′- TCAACAGCAACTCCCACTCTTCCA -3′ (forward) and 5′- TTGTCATTGAGAGCAATGCCAGCC -3′ (reverse).

### 4.8. Statistical Analysis

The data are presented as mean ± standard deviation (SD). A paired *t*-test was conducted, with a *p*-value of less than 0.05 considered statistically significant.

## 5. Conclusions

This study underscores the complexities of fusion protein production and exemplifies a rational protein design approach. By optimizing the linker sequence between RBP and IIIA and improving the stability of the middle domain (IIIA), we enhanced the overall structural stability as well as productivity, thermal stability, and bio-efficacy.

## 6. Patents

The experimental findings have been submitted for patent (10-2024-0114335) to the South Korea Patent Office.

## Figures and Tables

**Figure 1 ijms-26-00137-f001:**
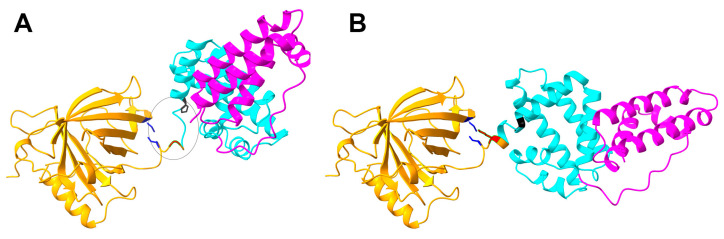
The predicted structures of R31 (**A**) and R31-G176K (**B**) generated using AlphaFold2. In these models, the RBP domain is shown in orange and the albumin domains IIIA and IB are depicted in cyan and magenta, respectively. The region linking the RBP and IIIA domains is marked with a black circle (**A**). Specific residues are highlighted as follows: C70 and C174 in blue, P182 in black, and 176 in red.

**Figure 2 ijms-26-00137-f002:**
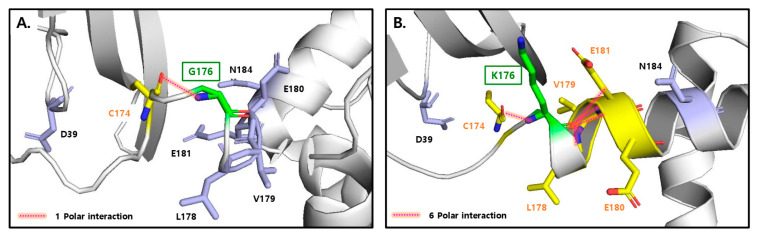
The AlphaFold2-predicted structure of the wild-type residue G176 (**A**) and the mutant K176 residue (**B**), with interaction analysis conducted using the Schrödinger Suite Release 2024-2. In these structures, G176K is highlighted in green, interacting residues in yellow, and surrounding residues in light blue. In both panels, oxygen and nitrogen atoms are marked in red and blue, respectively. Polar interactions are represented by red dashed lines.

**Figure 3 ijms-26-00137-f003:**
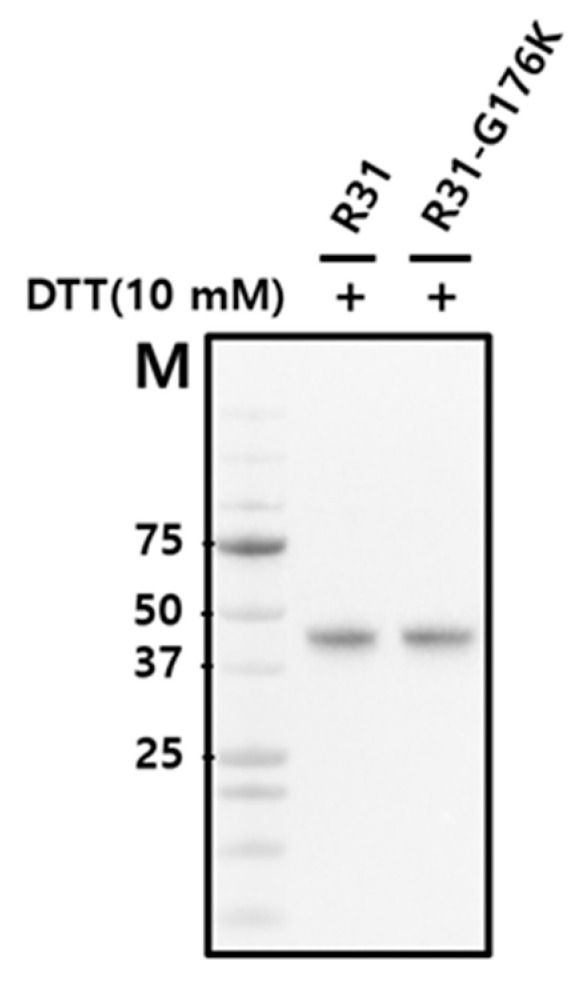
Expression level of the fusion proteins R31 and R31-G176K in Expi293 cells. Expi293 cells were transiently transfected with a plasmid encoding the fusion protein R31 and R31-G176K. The culture supernatant was analyzed by SDS-PAGE with 10 mM DTT and then subjected to Western blotting using an anti-His tag antibody. M denotes the molecular weight marker.

**Figure 4 ijms-26-00137-f004:**
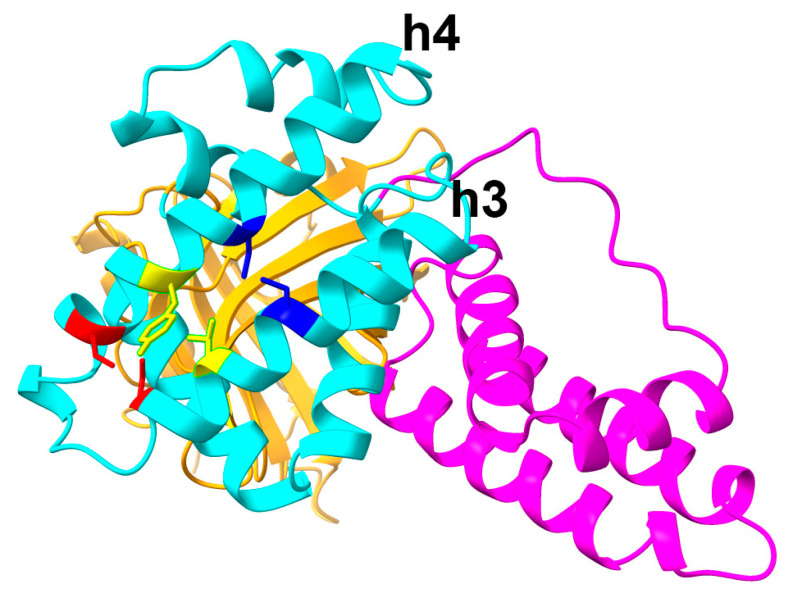
The predicted structure of R31 generated using AlphaFold2. In this model, the RBP domain is shown in orange and the albumin domains IIIA and IB are depicted in cyan and magenta, respectively. The native C235/C246 bond and N227C/V254C are colored in red and blue, respectively. Specific residues V231 and Y250 are highlighted in yellow.

**Figure 5 ijms-26-00137-f005:**
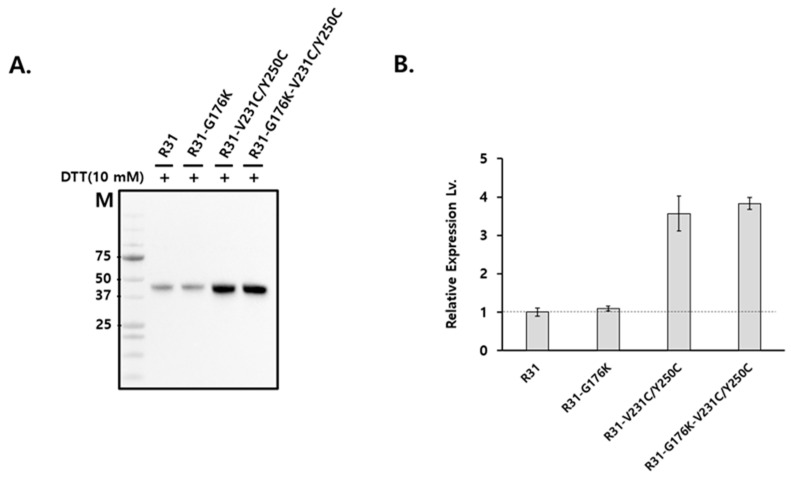
Comparison of expression levels of R31 variants in Expi293 cells. (**A**) Expi293 cells were transiently transfected with a plasmid encoding the fusion proteins R31, R31-G176K, R31-V231C/Y250C, and R31-G176K-V231C/Y250C. The culture supernatant was analyzed by SDS-PAGE with 10 mM DTT and then subjected to Western blotting using an anti-His tag antibody. M denotes the molecular weight marker. (**B**) Western blot results were quantified using ImageJ (version 1.53m). The quantitative densitometric data represent the means ± SD from two independent experiments.

**Figure 6 ijms-26-00137-f006:**
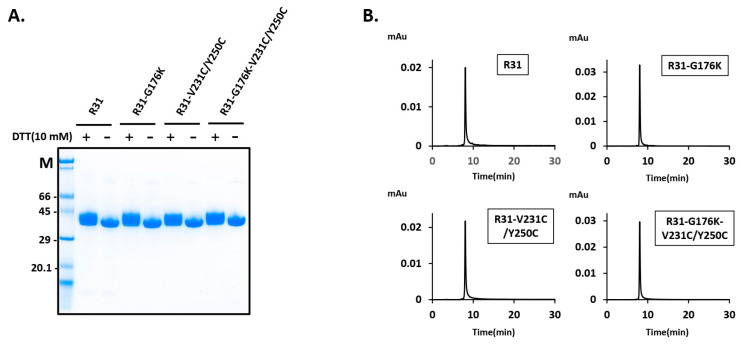
Expression and purification of R31 variants in Expi293 cells. Expi293 cells were transiently transfected with a plasmid encoding the fusion proteins R31, R31-G176K, R31-V231C/Y250C, and R31-G176K-V231C/Y250C. The resulting fusion proteins were purified using Ni Sepharose, followed by size exclusion chromatography. (**A**) SDS-PAGE analysis of the purified proteins, with (+) and without (-) 10 mM DTT, shown in the panel. M denotes the molecular weight marker. (**B**) The size exclusion chromatography profiles of the purified proteins.

**Figure 7 ijms-26-00137-f007:**
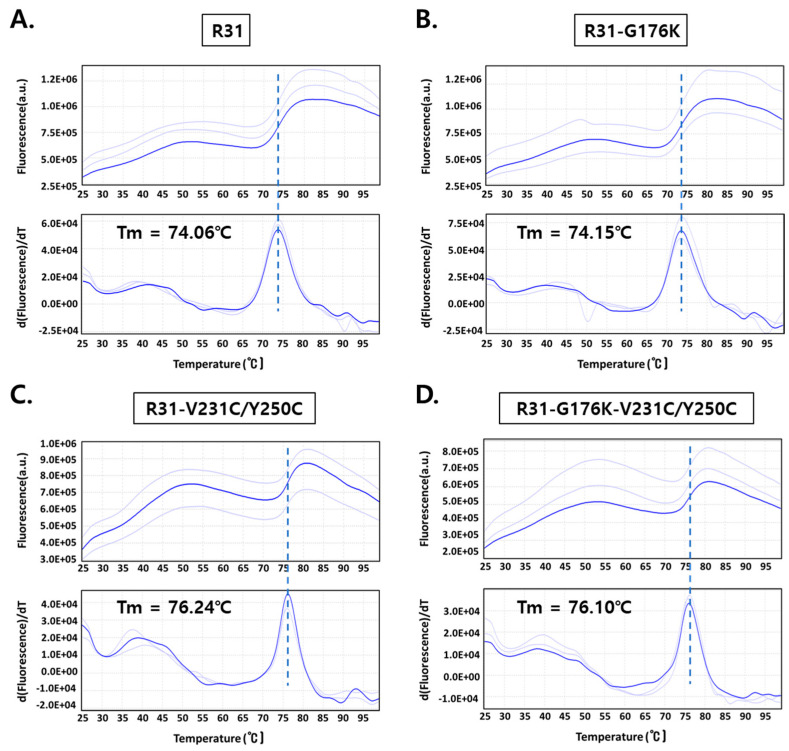
Protein thermal shift (PTS) assay results for R31 variants, R31 (**A**), R31-G176K (**B**), R31-V231C/Y250C (**C**), and R31-G176K-V231C/Y250C (**D**), in PBS (pH 7.4). The melting temperature (Tm) for each variant is as follows: R31 (74.06 °C), R31-G176K (74.15 °C), R31-V231C/Y250C (76.24 °C), and R31-G176K-V231C/Y250C (76.10 °C). The different shades of blue represent the results from three independent experimental replicates, while the blue line indicates the graph derived from the averaged values. The central dotted line represents the calculated thermostability temperature.

**Figure 8 ijms-26-00137-f008:**
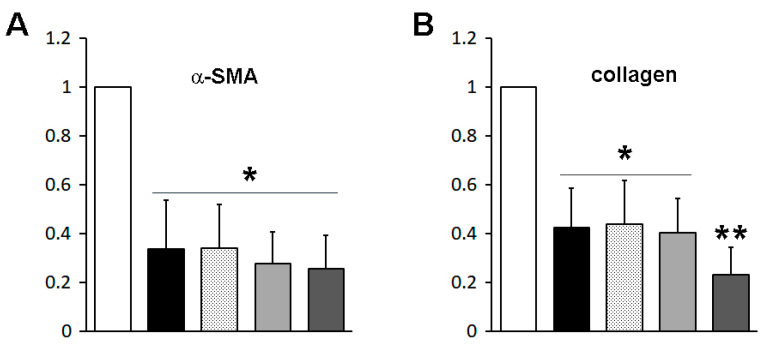
Anti-fibrotic effects of fusion proteins on hepatic stellate cells (HSCs). HSCs after passage 1 were treated with purified fusion proteins (0.75 μM), R31 (black), R31-G176K (hatched), R31-V231C/Y250C (light grey), or R31-G176K-V231C/Y250C (dark gray), for 16 h. The expression levels of alpha-smooth muscle actin (α-SMA) (**A**) and collagen type I (**B**) were assessed using real-time PCR. Statistical significance was determined by paired *t*-test (n = 3), and values are indicated as significant at * *p* < 0.05 or ** *p* < 0.01 compared to the untreated control cells.

**Figure 9 ijms-26-00137-f009:**
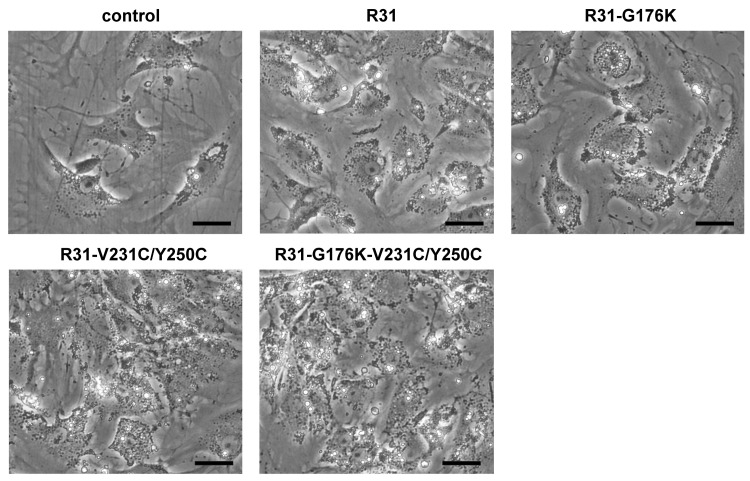
Fusion proteins induced phenotypic changes in hepatic stellate cells (HSCs). HSCs after passage 1 were treated with purified fusion proteins (0.75 μM), R31, R31-G176K, R31-V231C/Y250C, or R31-G176K-V231C/Y250C, for 16 h. Morphological changes were then assessed using a light microscope. Scale bar = 30 μm.

**Table 1 ijms-26-00137-t001:** The productivity of the fusion proteins in Expi293 cells.

Fusion Proteins	Productivity ^1^ (mg L^−1^)
R31	36.52 (±23.64)
R31-G176K	31.22 (±24.35)
R31-V231/Y250C	115.80 (±8.20)
R31-G176K-V231/Y250C	135.92 (±19.54)

Averages of three independent experiments; values in parentheses, ±SD.
^1^ Calculated from protein yields purified from 100 mL cultures.

## Data Availability

The data presented in this study are available on request from the corresponding author.

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
