# Peer review of "Strategic Optimization of the Middle Domain IIIA in RBP-Albumin IIIA-IB Fusion Protein to Enhance Productivity and Thermostability"

_ijms, 2024, doi:10.3390/ijms26010137_

Round 1

Reviewer 1 Report

Comments and Suggestions for Authors

This manuscript focuses on the fusion protein, involving albumin and RBP to treat tissue fibrosis. Fusion proteins is a growing market in the pharameutical industry, and there is always the need to increase the effectivness of these drugs. In previous reserach the authors showed that introducing a cysteine bridge in one place in the protein increases its thermal stability and functionallity. Herein, the authors succeeded to find another mutation to increase this stability and activity of the protein based on understanding the protein structure. The manuscript is well written, and it seems that it is also well conducted. I therefore recomend accepting it in its current version. 

Author Response

Comment 1: This manuscript focuses on the fusion protein, involving albumin and RBP to treat tissue fibrosis. Fusion proteins is a growing market in the pharameutical industry, and there is always the need to increase the effectivness of these drugs. In previous reserach the authors showed that introducing a cysteine bridge in one place in the protein increases its thermal stability and functionallity. Herein, the authors succeeded to find another mutation to increase this stability and activity of the protein based on understanding the protein structure. The manuscript is well written, and it seems that it is also well conducted. I therefore recomend accepting it in its current version. 

Response 1: I am grateful for the positive and encouraging feedback provided by the reviewer.

Reviewer 2 Report

Comments and Suggestions for Authors

In this manuscript the authors optimized the linker sequence (middle domain IIIA) of a previously developed fusion protein named R31, which combined a retinol-binding protein (RBP) with albumin domains IIIA and IB, linked by a sequence (AAAA), and includes an additional disulfide bond (N227C-V254C) in IIIA. This fusion protein effectively inhibited hepatic stellate cell activation, which is connected with tissue fibrosis. Their optimization resulted in increased productivity and thermal stability.

The research are well conducted and described. However I found some inconsistencies.

Lines 81 and 95 – please add also the reference to Fig.2 (except the ref to the Fig1).

Line 180 – figure 6 caption - there is the indication of performing SDS-PAGE analysis with and without 10mM DTT. However nowhere in the main text (especially in  results section) it is not comment   this addition of DTT. Please add the explanation.

Line 194 – “…on activated HSCs in vitro.” Please add in this place, that this cells were obtained from mice as described in Materials and Methods.

Line 242 – please add the ref to the supplementary Fig. S1, where schematic representation of the fusion proteins is presented.

Line 295 – please add the composition of the 5xreducing loading buffer.

Therefore I recommend minor revision.

Author Response

Comment 1: Lines 81 and 95 – please add also the reference to Fig.2 (except the ref to the Fig1).

Response 1: As per your request, we have added “Figure2” on lines 81 and 95.

Comment 2: Line 180 – figure 6 caption - there is the indication of performing SDS-PAGE analysis with and without 10mM DTT. However nowhere in the main text (especially in  results section) it is not comment   this addition of DTT. Please add the explanation.

Response 2: In response to your request, we have added the explanation for the SDS-PAGE results with and without DTT on lines 162-166.

Comment 3: Line 194 – “…on activated HSCs in vitro.” Please add in this place, that this cells were obtained from mice as described in Materials and Methods.

Response 3: As per your request, we have included the phrase on line 197.

Comment 4: Line 242 – please add the ref to the supplementary Fig. S1, where schematic representation of the fusion proteins is presented.

Response 4:  As per your request, we have added the reference to the Supplementary Figure S1 on lines 71, 288, and 359.

Comment 5: Line 295 – please add the composition of the 5xreducing loading buffer.

Response 5: We mistakenly provided an incorrect description of the reducing loading buffer, but we have now corrected it and given an accurate description on lines 299-301.